# Shear Wave Elastography Implementation on a Portable Research Ultrasound System: Initial Results

Damian Cacko [1,2,*] and Marcin Lewandowski [1,2,*]

1    Institute of Fundamental Technological Research, Polish Academy of Sciences, 02-106 Warsaw, Poland
2    us4us Ltd., 02-106 Warsaw, Poland
*    Correspondence: damian.cacko@us4us.eu (D.C.); marcin@us4us.eu (M.L.)

**Abstract:** Ultrasound shear wave elastography (SWE) has emerged as a promising technique that enables the quantitative estimation of soft tissue stiffness. However, its practical implementation is complicated and presents a number of engineering challenges, including high-energy burst transmission, high-frame rate data acquisition and high computational requirements to process huge datasets. Therefore, to date, SWE has only been available for high-end commercial systems or bulk and expensive research platforms. In this work, we present a low-cost, portable and fully configurable 256-channel research system that is able to implement various SWE techniques. We evaluated its transmit capabilities using various push beam patterns and developed algorithms for the reconstruction of tissue stiffness maps. Three different push beam generation methods were evaluated in both homogeneous and heterogeneous experiments using an industry-standard elastography phantom. The results showed that it is possible to implement the SWE modality using a portable and cost-optimized system without significant image quality losses.

**Keywords:** ultrasound imaging; shear wave elastography; high-frame rate imaging; medical system design





## 1. Introduction

Since its beginnings in the early 1990s [1], ultrasound-based elastography has developed rapidly. The technique can be used for the characterization of the mechanical properties of tissues and provides diagnostically relevant information that cannot be obtained with standard B-mode ultrasound imaging.

### 1.1. Shear Wave Elastography Principles

Out of all of the elastography approaches that have been developed so far, shear wave elastography (SWE) has emerged as a promising technique of growing interest. In strain elatography (a longstanding and well-established method), tissue stiffness maps are obtained through an analysis of the strains within the tissue under stress, which is applied by the operator. It is typically conducted by pushing the ultrasound probe against the body. In contrast, the SWE technique relies on remotely inducing tissue displacement using the acoustic radiation force (ARF), as first proposed by Sarvazyan et al. [2]. The ARF is generated by an ultrasound beam (also called a push beam or push pulse) and is defined in Equation (1) as follows:

$$\vec{F} = \frac{2\alpha \vec{I}}{c} \tag{1}$$

where $\vec{F}$ is the generated force, $\alpha$ is the absorption coefficient, $I$ is the acoustic intensity and $c$ is the speed of the longitudinal wave propagation. As a result of the tissue displacements from applying the ARF, transient shear waves that propagate in the perpendicular direction to the ARF are induced in the elastic medium, as shown in Figure 1. Assuming that soft tissues are incompressible, isotropic, linear and elastic, a local shear modulus $\mu$ can be

obtained by finding the local shear wave propagation speed $c_s$, namely $\mu = \rho \cdot c_s^2$, where $\rho$ is the density ($\approx 1000 \, \text{kg m}^{-3}$ for soft tissue). As for soft tissue, the bulk modulus $B$ is much higher than the shear modulus ($B >> \mu$) and the Young's modulus $E$ of the medium can be estimated using:

$$E = 3\mu = 3\rho \cdot c_s^2 \tag{2}$$

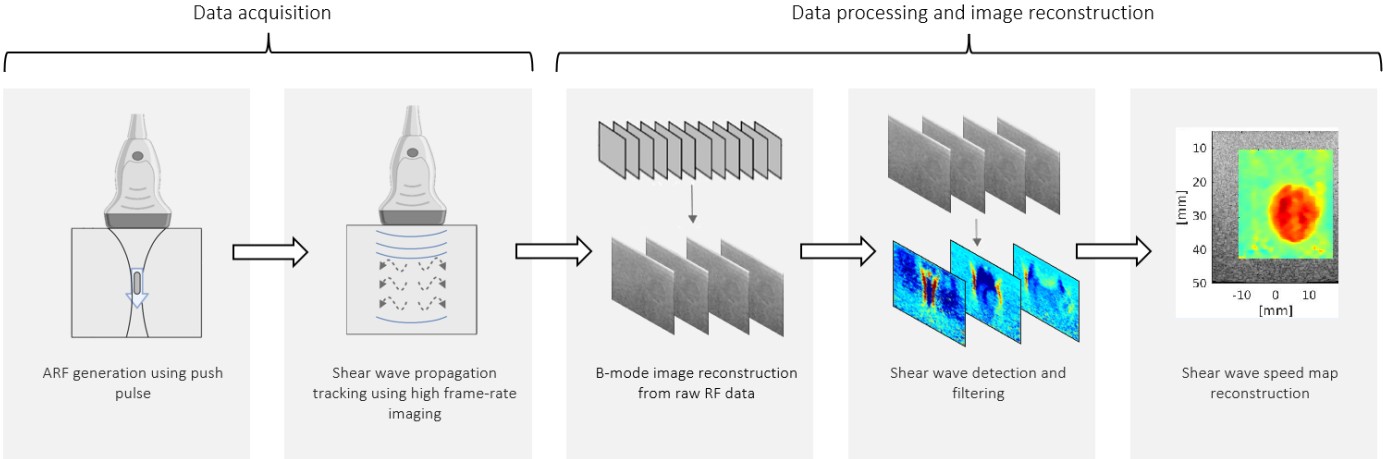

**Figure 1.** A general measurement procedure of the SWE technique.

A general SWE measurement procedure is shown in Figure 1. Just as in the vast majority of ultrasonic techniques, the process consists of two stages: data acquisition, followed by data processing with image reconstruction. First, the ARF is generated in the medium using a push pulse. The shear wave motions that result from the impulsive excitation of the ARF can be tracked ultrasonically by detecting small axial displacements along the shear wave propagation path using pulse–echo ultrasound. The shear wave speed (SWS) depends on the characteristics of the elastic medium. Typically, high-frame rate plane wave imaging (PWI) must be used for effective shear wave tracking. Immediately after the end of the push pulse generation, the scanner switches into a high-frame rate imaging mode and captures the RF data. Then, the RF data are used in the beamforming process to reconstruct a series of images. Successive beamformed images are the inputs for the correlation-based algorithm that performs shear wave detection by obtaining the particle axial displacement of each pixel. The known approaches are described in detail in [3]. Finally, shear wave motion data are used to obtain a tissue stiffness map. Several algorithms have been proposed for this, most of which are based on time-of-flight (ToF) methods. These approaches can be divided into two types: linear regression methods [4–9] and shear wave trajectory detection methods [10,11].

The key advantage of SWE is that external, operator-dependent stress is not required due to use of the ARF. Moreover, whereas strain elastography only provides quantitative soft tissue stiffness maps, SWE can estimate soft tissue stiffness both qualitatively and quantitatively. It also possesses the common features of ultrasound imaging, i.e., it is non-invasive, fast and relatively low-cost and it has wide applications, such as [12] fibrosis assessment for chronic liver disease, breast cancer screening, thyroid nodules assessment, gastrointestinal wall diagnostics, prostate abnormality screening and cardiovascular system diagnostics [13].

### 1.2. Background and Motivation

A substantial improvement has been achieved in the field of SWE in last two decades and many approaches for SWE implementation have been reported by various research groups. Each method differs with regard to push beam generation, data acquisition scheme, shear wave tracking methods and the SWS reconstruction algorithm that are used. In [14], Nightingale et al. used a single focused push beam to generate the ARF. However, as a

result of focusing, the shear waves were generated in close proximity to the focal point of the push beam. Therefore, high-quality elasticity map reconstruction was only possible within a limited depth-of-field (DOF). To address this issue, a new push beam generation method was proposed by Bercoff et al. in [15], called supersonic shear imaging (SSI). In SSI, a sequence of beams that are focused at consecutive depths is used to generate a broad shear wavefront over a high axial extent to cover an extended field of view (FOV). In addition, an imaging rate of several thousand frames per second (FPS) is utilized in this method to track the shear waves. It allows for the capture of real-time shear wave propagation within the whole FOV using a single acquisition scheme, which has been introduced by the same group before in [16]. Although a single SSI acquisition covers a large part of the FOV, there is still a lack of shear waves in the beam axis. Therefore, the authors proposed the use of three acquisitions, each with a different push beam lateral position [4]. An alternative approach was proposed by another group in a series of works [7–9], in which multiple beams that are arranged in a comb pattern are used to generate shear waves. As the comb push beam-induced shear waves propagate within the whole FOV, a comprehensive elasticity map is reconstructed by applying only one acquisition. Nevertheless, the unfocused comb push beams feature a low penetration in depth, while the focused beams have limited DOF. Recent developments have shown that even more complex beam patterns, such as Bessel beams, can be used for a better optimized shear wave energy distribution within the FOV and improved power efficiency [17].

The practical implementation of the SWE technique is a complex engineering task and poses a number of technical challenges. Firstly, the generation of the ARF involves the transmission of high-energy and long transmit sequences using a wide aperture (many probe elements driven simultaneously). Consequently, it requires robust transmit capabilities in the ultrasound system and can cause significant heat dissipation in the driving electronics and the probe itself. Secondly, shear wave propagation tracking requires high-frame rate imaging (a minimum of 1000 FPS and typically >5000 FPS). This leads to a significant amount of data being captured and transferred from the ultrasound system to a PC for further processing. The huge dataset to be processed by the software and the complex nature of the elastography reconstruction algorithms demand high computational performance in order to reconstruct elasticity maps, especially in real-time applications.

Two-dimensional SWE has been successfully developed and commercialized by several leading companies, including the Philips (Amsterdam, The Netherlands) EPIQ series and Affiniti 70, the SuperSonic Imagine (Aix-en-Provence, France) Aixplorer and MACH series, the Canon Medical Systems (Otawara, Japan) Aplio series, the Siemens (Erlanden, Germany) ACUSON series and the General Electric (Boston, MA, USA) LOGIQ series. Nevertheless, these devices are all advanced clinical cart scanners with proprietary processing algorithms and thus, are not useful for research purposes. To facilitate the development and evaluation of new diagnostic ultrasound methods and algorithms, researchers require a dedicated system that features the full configurability of transmit patterns and acquisition sequences, provides access to raw echo prebeamformed received data (RF data) and has sufficient computational resources to allow for real-time imaging implementation [18]. In addition, to support the SWE technique, research platforms should also allow for the generation of long transmit bursts for push beam generation. To date, the Vantage system with an "extended transmit option" (Verasonics, Kirkland, WA, USA) is the only research scanner available that features all of these functionalities. However, this scanner is bulky and expensive.

In this work, we present a dedicated high-power TX add-on board (TXPB-256) for a portable and low-cost ultrasound research platform (us4R-lite), which enables the implementation of the SWE technique. In the experimental part of the study, the system performance of the shear wave speed measurement was validated using an industry-standard elastography phantom and various push beam generation techniques. The quality of the resulting stiffness map images was also evaluated. The obtained results showed that SWE implementation is feasible on a portable and limited hardware platform.

## 2. Materials and Methods

### 2.1. Ultrasound System Architecture

The architecture of the us4R-lite (us4us Ltd., Warsaw, Poland) research platform is depicted in Figure 2. The system architecture was optimized for the implementation of the software-defined ultrasound paradigm by featuring high-speed data acquisition and streaming.

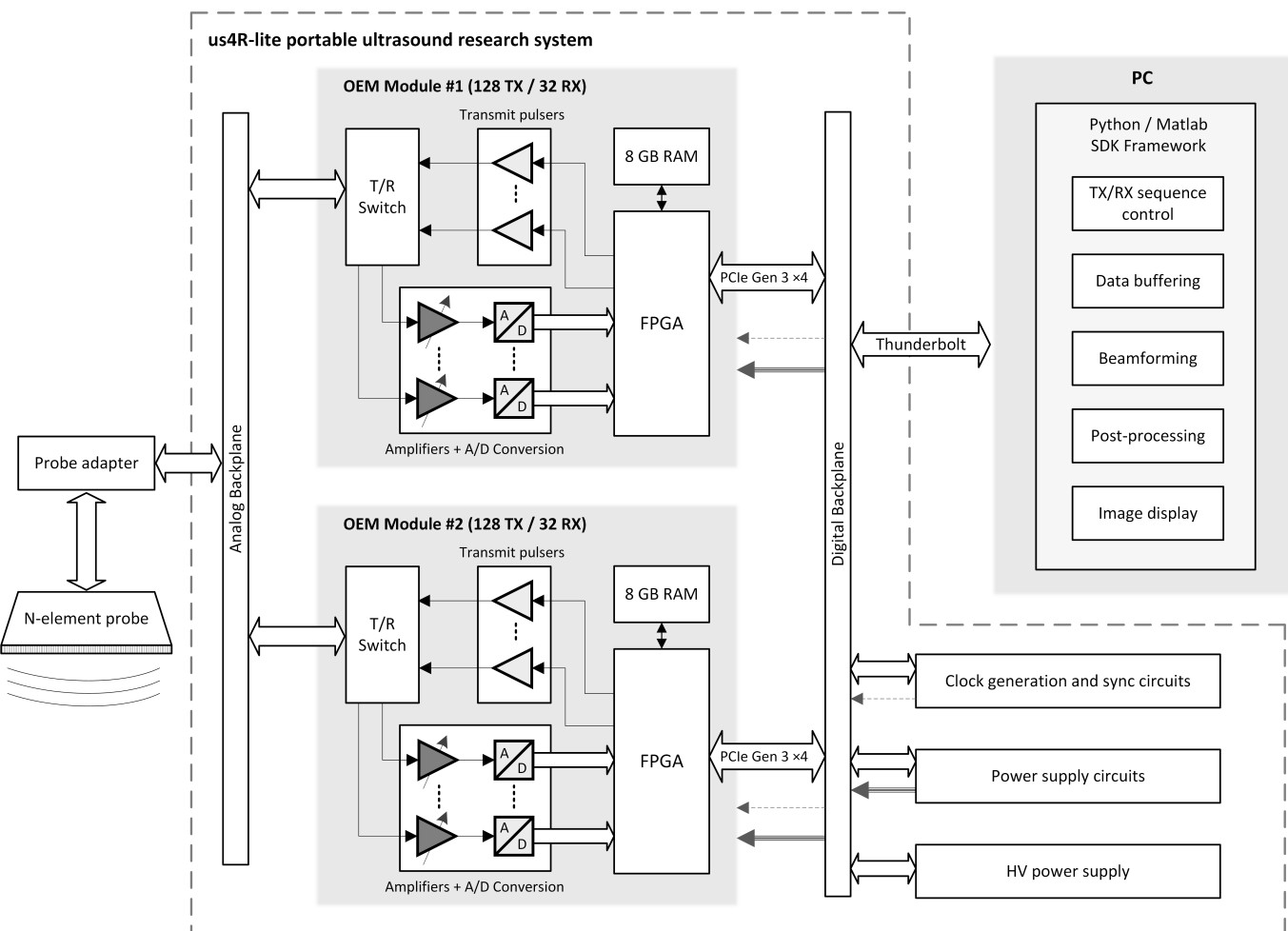

**Figure 2.** A simplified system architecture of the us4R-lite portable ultrasound research platform. Some connections are hidden for clarity. Abbreviations: T/R, transmit/receive; A/D, analog to digital; FPGA, field-programmable gate array; SDK, software development kit; HV, high voltage.

The system was built around two us4OEM (us4us Ltd., Warsaw, Poland) ultrasound front-end modules, each containing both transmit (TX) and receive (RX) electronics. Each module has individually controlled 128 TX and 32 RX channels, totaling 256 TX and 64 RX channels for the us4R-lite system. The transmitter part is based on 3-level transmit pulser-integrated circuits that drive individual probe elements with high-voltage (HV) bipolar square waves. The receiving path consists of an amplifying stage (a low-noise amplifier and time-gain control amplifier), followed by an analog to digital converter (ADC). The transmit and receive paths are connected in the transmit/receive (T/R) switch, which connects the transducer elements to either the transmitter or receiver subsystem via multiplexing. Both the TX and RX parts are controlled by a field-programmable gate array (FPGA) that is configured with custom firmware. The FPGA is also responsible for the execution of predefined TX/RX acquisition sequences, which need to be programmed by the software prior to the acquisition. For TX operation, the FPGA generates the transmit patterns that are to be amplified by the TX pulsers. On the RX side, the FPGA captures the output data

streams from the ADCs, which are buffered in the off-chip DRAM. As there is no integrated hardware-based data processing, all of the acquired RF data are fed into the PC-based back-end computing. The lack of demodulation and beamforming results in a large volume of data being transferred to the PC, especially in real-time applications [19]. For this reason, the data in the us4R-lite system is streamed to the PC over a high-speed Thunderbolt-3 or PCIe 3.0 interface with an effective bandwidth of 2.5–6 GB/s.

The other components that comprise the system include clock generation and synchronization circuits, a power supply (which generates all of the required voltage levels for the other subsystems) and a HV power supply (which generates high-voltage bipolar waves for the transducer excitation). All of the subsystems are connected by digital and analog backplanes. The probe adapter is an exchangeable interface that supports various ultrasound probes.

Despite the promising results of the system evaluation of push pulse generation in the previous work by our team [20], multiple TX pulsers were damaged during the preliminary SWE test runs due to HV overshoots or overheating. Therefore, in order to increase the robustness of our system's transmit capabilities, we designed a dedicated 256-channel transmitter subsystem, called the transmit push beamformer board (TXPB-256). Taking the form of an extension board that replaces the analog backplane (see Figure 2), this system supersedes the transmitters and T/R switches on the us4OEM boards. The architecture of the TXPB-256 board is shown in Figure 3.

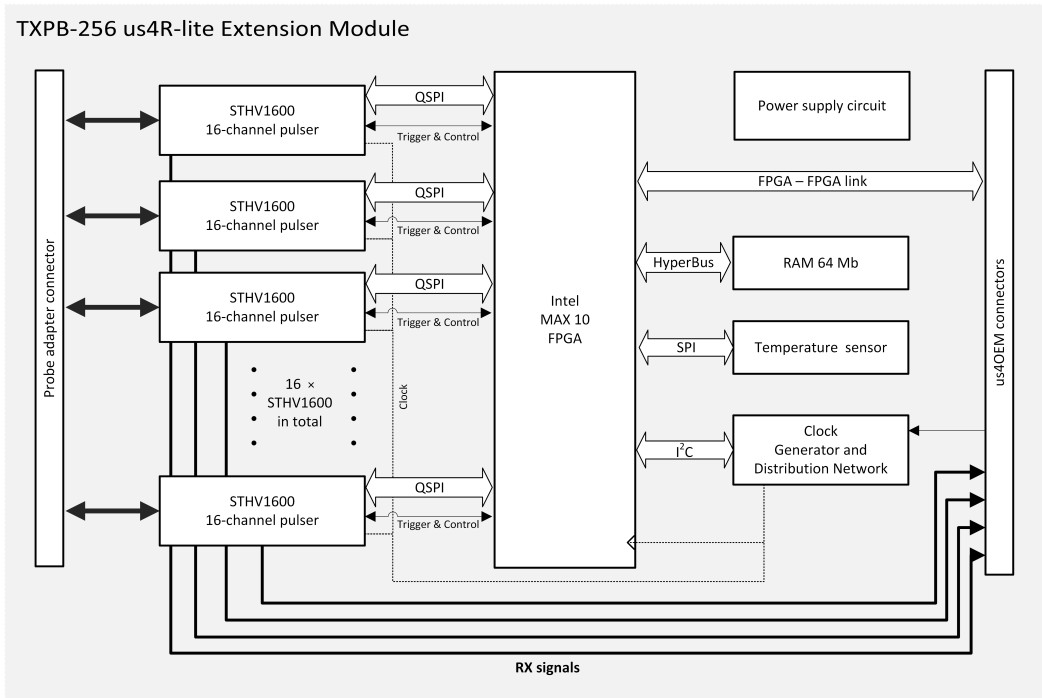

**Figure 3.** The architecture of the transmit push beamformer (TXPB-256) extension board.

The TXPB-256 utilizes new generation 16-channel STHV1600 transmit pulser ICs (STMicroelectronics, Geneva, Switzerland). Each of these pulsers consists of analog and digital parts. The analog part implements the high-voltage output drive and T/R switch functionality. The digital part consists of a digital transmit beamformer and a set of registers to configure the transmit aperture, transmit delay profile, output waveforms and other minor settings. As the pulser has a memory that is only sufficient for storing a single TX definition, we implemented a controller using MAX 10 FPGA (Intel, Santa Clara, CA, USA), which reloads the definitions in each pulser prior to every transmit event via QSPIs (quad serial peripheral interfaces). The TX sequence and TX-related definitions are easily configured by the software and are stored in the on-board memory. A summary of the transmit capabilities is presented in Table 1.

**Table 1.** The transmit capabilities of the us4R-lite research platform with the (TXPB-256) extension module.

| Feature | Value |
|---|---|
| TX channels | 256 |
| HV transmit voltage | Configurable, up to 180 Vpp |
| Output current per channel | 2 A or 4 A |
| Transmit frequency range | 0.1–50 MHz [1] |
| Transmit aperture | Arbitrary |
| Transmit pulse capability | Arbitrary 3-level square wave pattern for each channel |
| Transmit apodization | Yes [2] |
| Transmit delay | 0–327 µs |
| Timing resolution | 5 ns |
| Theoretical maximum PRF | 100 kHz |

[1] Maximum frequency is load-dependent; [2] implemented using pulse width modulation.

## 2.2. Push Beam Generation

The us4R-lite research platform with the TXPB-256 extension module, as described above, was used in this work to generate various push beams alongside a 128-element L7-4 linear array probe (Philips Healthcare, Amsterdam, The Netherlands) with a center frequency of 5 MHz and 0.3 mm of pitch. Three different push beams generation methods were used, all of which are presented schematically in Figure 4.

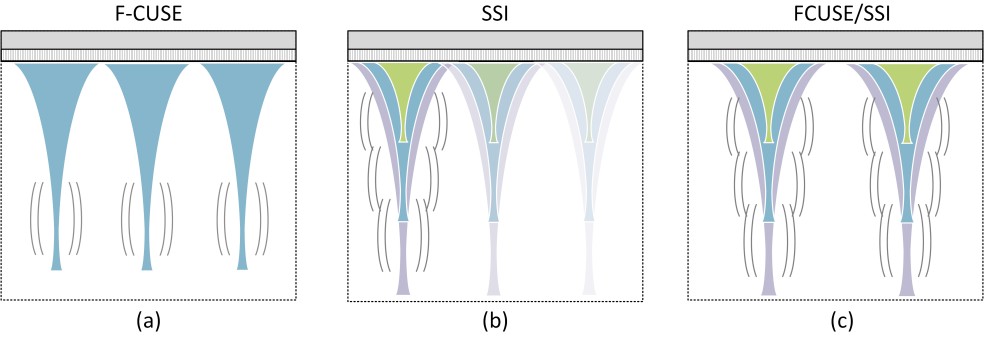

**Figure 4.** The push beam patterns that were used in this study: (**a**) FCUSE; (**b**) SSI; (**c**) FCUSE-SSI.

The details of each method were as follows.

- FCUSE: The first method followed the principle that was proposed by Song et al. [21]. The transducer aperture was divided into three groups of 42 elements, each simultaneously producing a focused beam at a depth of 30 mm ($F/\# = 2.4$). The push length was 800 µs.
- SSI: The second method followed the principle that was proposed by Tanter et al. in [4]. Three acquisitions were used in total, each producing three laterally aligned beams that were focused at successive depths, which were generated sequentially by increasing the aperture size along with focusing the depth to maintain a constant f-number. The consecutive beams used 24, 50 and 66 elements and were focused at depths of 15 mm, 30 mm and 40 mm, respectively (constant $F/\# = 2$). The single push length was 80 µs.
- FCUSE-SSI: In the third method, we proposed a combination of the two previous methods. Two focused beams were generated simultaneously, as in FCUSE, and then the beams were focused at consecutive depths and generated sequentially, as in SSI. It was equivalent to the generation of one FCUSE beam after another, increasing the focal point of the beam in each iteration. The consecutive beams used 24, 50 and 64 elements and were focused at depths of 15 mm, 30 mm and 40 mm and resulted in $F/\#$ values of 2, 2 and 2.1, respectively. The single push length was 120 µs.

A push pulse frequency of 4.4 MHz was used for all experiments. For a fair comparison of each method, the total transmitted energy for all schemes was kept at the same level. The energy transmitted from a single transducer element was obtained as in [22]:

$$E = \frac{V_{rms}^2}{Z} \cdot T \tag{3}$$

where $V_{rms}$ is the root mean square of the excitation voltage waveform, $Z$ is the transducer element impedance and $T$ is the push pulse duration. The total energy is equal to the sum of the transmitted energy from all of the individual elements in all acquisitions. Because the same probe was used at the same center frequency of the push pulse in all cases, $Z$ was assumed to be constant, as well as $V_{rms}$. Thus, only the total number of active elements and the total push duration could be used to compare the transmitted energies.

### 2.3. Data Acquisition

The data acquisition scheme that was used in the experiments is shown in Figure 5. The measurement procedure started with the generation of the push beams. In the cases of the SSI and FCUSE-SSI methods, consecutive beams were generated with a pulse repetition interval (PRI) that was equal to the push beam duration of +20 μs. As detailed in the previous section, in SWE, the push beam generation is followed by shear wave tracking. Immediately after the end of the push beam generation, the us4R-lite scanner was switched to the high-frame rate PWI mode, transmitting two cycles of a center frequency of 4.4 MHz with a pulse repetition frequency (PRF) of 10 kHz and using all of the probe elements to capture echoes from depths of up to 50 mm. In order to improve the B-mode image quality, a coherent plane wave imaging method (CPWI) with angle compounding, as proposed by Montaldo et al. in [23], was utilized in this study. The set of angles used was $[-4°, 0°, 4°]$. In each acquisition, a total of 150 frames was captured. Due to the limited number of RX channels (64) in the us4R-lite system, each transmission was repeated twice to capture the whole 128-element receiving aperture, which reduced the effective frame rate twice (down to 5 kHz) and required 300 plane wave transmissions per acquisition.

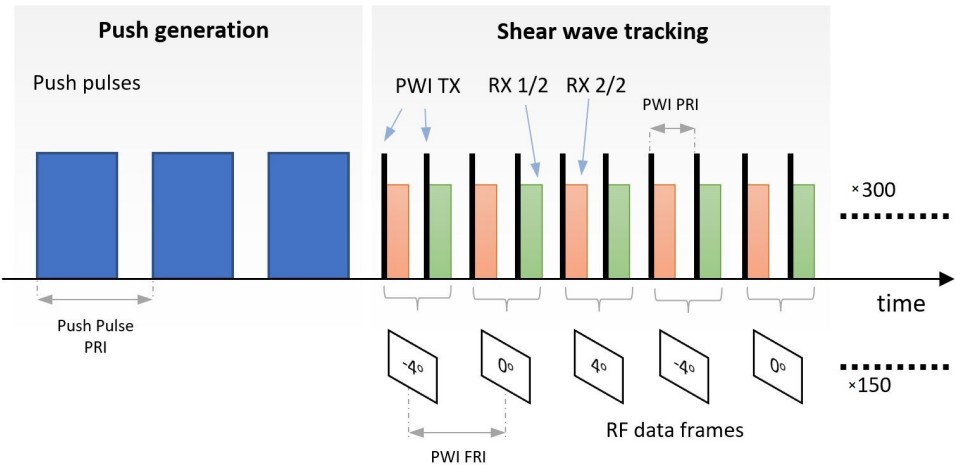

**Figure 5.** An example data acquisition scheme for the FCUSE-SSI method. To capture the RF data for a full receiving aperture, two transmissions were required, each followed by a 64 RX aperture size. Abbreviations: PRI, pulse repetition interval; FRI, frame repetition interval.

### 2.4. Stiffness Map Reconstruction Algorithm

The data obtained during the acquisition were used as the inputs for a reconstruction algorithm that consisted of multiple steps. Firstly, to improve the signal to noise ratio (SNR) of the RF data, a finite impulse response (FIR) filter was used to filter out the components outside of the 4 MHz–7 MHz band of the probe. Then, standard PWI beamforming was performed, which produced a beamformed in-phase/quadrature (IQ) frame for each angle

transmission with a pixel size of $0.1\,\text{mm} \times 0.1\,\text{mm}$. As angle compounding was utilized, an initial phase delay compensation was performed using the method that was proposed by Lee et al. [24]. Once this was accomplished, the angle compounding was performed using a sliding average technique to maintain the high frame rate.

The angle-compounded beamformed data were then used for the shear wave motion detection. Shear wave motion produces small tissue displacements that result in differences in the arrival times of the corresponding image line echoes from one frame to another. In the case of IQ data, shear wave motion manifests as a phase difference $\delta\varphi$ in the corresponding pixels of consecutive frames. The detection of this phase shift allows us to obtain the local axial velocity $v_z$:

$$v_z = \frac{c \cdot \delta\varphi}{4\pi f_c \cdot FRI} \tag{4}$$

where $c$ is the speed of sound (average of $1540\,\text{m/s}$ in soft tissue), $f_c$ is the tracking pulse center frequency and FRI is the frame repetition interval. In this work, the Kasai 1D-autocorrelator algorithm [25] was used to calculate the $\delta\varphi$ for each pixel in each frame. This method measures the average phase shift over axial sample ranges $M$ and $N$ in a slow-time domain. In this study, the values of $N = 2$ and $M = 8$ were used to obtain the 3D shear wave motion data $v_z(x, z, T)$, which were the transient axial velocity as a function of the axial position $z$, lateral position $x$ and slow-time $T$. To reduce the noise content in the signals and the random errors in the shear wave motion data due to finite window lengths or decorrelation, low-pass filtering was applied in the slow-time domain with a cut-off frequency of $700\,\text{Hz}$.

The next step of the algorithm involved using a directional filter, which was similar to that proposed by Manduca et al. in [26]. As proposed in [27], a 2D filter was applied in the Fourier $(k_x, \omega)$ space. The motion data were sliced along the depth dimension $(z)$ to form 2D slices $v_z(x, T)$. A 2D FFT was used to transfer the data from each slice to the Fourier domain and the filtering was performed by multiplying the slices by masks that were designed to extract the shear waves that were propagating from left to right (LR) $v_{zLR}$ and right to left (RL) $v_{zRL}$. In addition, the masks were designed to filter out all data components that were related to shear waves propagation speeds that were outside of the $0.5$–$5.5\,\text{m/s}$ range. The mask edges were apodized to minimize the ripples in the spatio-temporal domain. Except separating shear waves that were traveling in opposite directions, these filters attenuated the wave reflections from the interferences of different stiffnesses, such as lesions or stiffer inclusions, which could possibly result in artifacts in the SWS estimates.

A time-of-flight (ToF) algorithm, which was similar to that proposed in [4,8], was used in this work to obtain the local SWS in each pixel. The algorithm was run twice to obtain the SWS estimates from both the LR and RL datasets. In this method, we calculated normalized cross-correlations of the shear wave axial velocity profiles along the lateral dimension $m$, which were separated by a lateral window size $p$ but were on the same depth $n$, to find the shear wave arrival delay $\Delta t$:

$$\Delta t(x, z) = \frac{1}{FRI} \cdot \left\{ \underset{j}{\arg\max}\, CC[v_z(x - p/2, z, T), v_z(x + p/2, z, T - j)] \right\} \tag{5}$$

To facilitate accurate cross-correlation results, the Tukey window was applied to the shear wave motion profiles and the profiles were interpolated by a factor of 20 before the cross-correlation was performed. This calculation was repeated to obtain the $\Delta t$ and $CC$ coefficient $(CC_c)$ pairs for each image pixel. The local SWS $v_{SW}$ could be obtained using a simple formula:

$$v_{SW}(x, z) = \frac{p \cdot \Delta x}{\Delta t(x, z)} \tag{6}$$

where $\Delta x$ is the pixel size.

The last step of the reconstruction algorithm involved composing a final SWS map $M_{SW}$ for each pixel $(x, z)$ by compounding the SWS estimates that were obtained from the $RL$ and $LR$ shear wave motion data $(V_{SW-RL})$ and $(V_{SW-RL})$. The center of the final SWS map contained contributions from both the $RL$ and $LR$ wave fields and was calculated as a weighted sum:

$$M_{SW}(x, z) = \frac{V_{SW-LR}(x, z) \cdot CC_{SW-LR}(x, z) + V_{SW-RL}(x, z) \cdot CC_{SW-RL}(x, z)}{CC_{SW-LR}(x, z) + CC_{SW-RL}(x, z)} \quad (7)$$

Because there was not an adequate amount of data to be used close to the final map borders due to the lack of shear wave presence, the left edge was only taken from the $RL$ map and the right edge was only taken from the $LR$ map. To achieve smooth transitions between all of the regions, a sigmoid-based data weighting was applied across the region boundaries, as proposed in [28].

### 2.5. Phantom Experiments and Quality Metrics

A complete test setup is presented in Figure 6. An evaluation of the system performance was carried out using a tissue-mimicking Model 049A elasticity Q/A phantom (CIRS, Norfolk, VA, USA). The phantom had four stepped cylinder mass targets of known stiffness and six diameters. Before performing the measurements, standard real-time B-mode imaging was used for probe placement over an inclusion. For all subsequent methods, the probe position was fixed using a handle, which was aimed at a given target. The phantom region with no inclusions was used for the homogeneous phantom experiments. To evaluate the performance of each push generation method (FCUSE, SSI and FCUSE-SSI) and for comparison purposes, a number of quality metrics were calculated. The shear wave motion energy in each pixel $SW_E(x, z)$ was calculated as:

$$SW_E(x, z) = \sum_{i=1}^{N} v_{zf}(x, z, i)^2 \quad (8)$$

where $N$ is the number of samples in the slow-time domain and $v_{zf}$ is the shear wave axial velocity after the frequency and directional filtering. The $SW_E(x, z)$ maps were calculated separately for both the $RL$ and $LR$ datasets and were then added together.

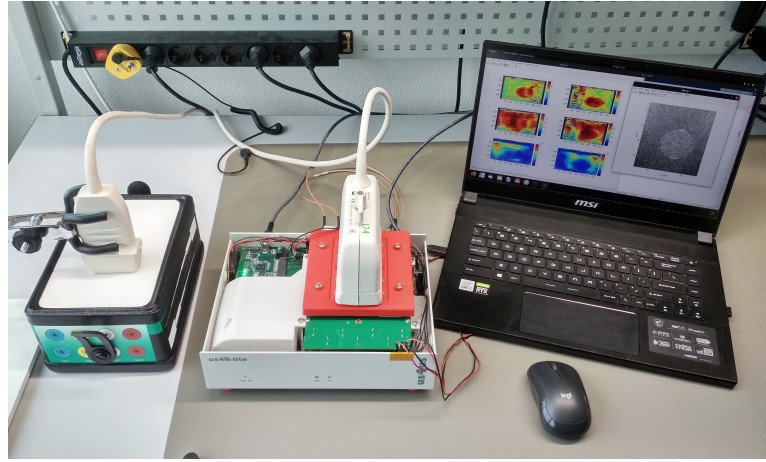

**Figure 6.** The setup that was used in this work. Left to right: the L7-4 probe over the CIRS 049A elastography phantom, the us4R-lite portable research platform with the TX extension board mounted and connected to the PC with Mathworks MATLAB software running.

In the homogeneous phantom experiment, the bias and SNR of the stiffness map was obtained as follows:

$$Bias = \frac{SWS_E - SWS_N}{SWS_N} \cdot 100\% \quad (9)$$

$$SNR_{dB} = 20log_{10}(\mu_{SWS}/\sigma_{SWS}) \tag{10}$$

where $SWS_E$ and $SWS_N$ are the estimated and nominal shear wave speed values, respectively, $\mu_{SWS}$ is the mean value and $\sigma_{SWS}$ is the standard deviation of the selected area of the stiffness map.

In the heterogeneous phantom experiments, the contrast to noise ratio (CNR) was derived using the mean SWS values of the inclusion and the background, $\mu_I$ and $\mu_B$, and the standard deviations of those values, $\sigma_I$ and $\sigma_B$, within manually extracted regions of the stiffness map:

$$CNR_{dB} = 20log_{10}\left(\frac{|\mu_I - \mu_B|}{\sqrt{\sigma_I^2 + \sigma_B^2}}\right) \tag{11}$$

## 3. Results

This section is organized as follows. The first part evaluates the developed system with regard to the feasibility of implementing the SWE technique by showing the results of the data processing at key stages of the reconstruction algorithm. The second part presents the results of the homogeneous phantom experiments using the three beam generation approaches. In the third part, the results of similar experiments with a heterogeneous phantom are presented.

### 3.1. System Validation

Figure 7 shows the reconstructed B-mode images of various elastography phantom regions, which were obtained from the plane wave acquisitions as described in Section 2.3. Although the full reconstruction of the B-mode images was not required for the SWS map reconstruction, an inspection of the B-mode images was performed to ensure proper probe placement. It can be observed that the inclusions of stiffness that differed from the background were all visible in the B-mode images. In all cases, the contrast was poor and almost the same for all of the inclusion types and diameters. Although the B-mode imaging did not provide any information that allowed us to distinguish the inclusion type, the inclusion shapes and diameters were reconstructed very well. Therefore, the B-mode images were used to define the ROIs in the heterogeneous experiments.

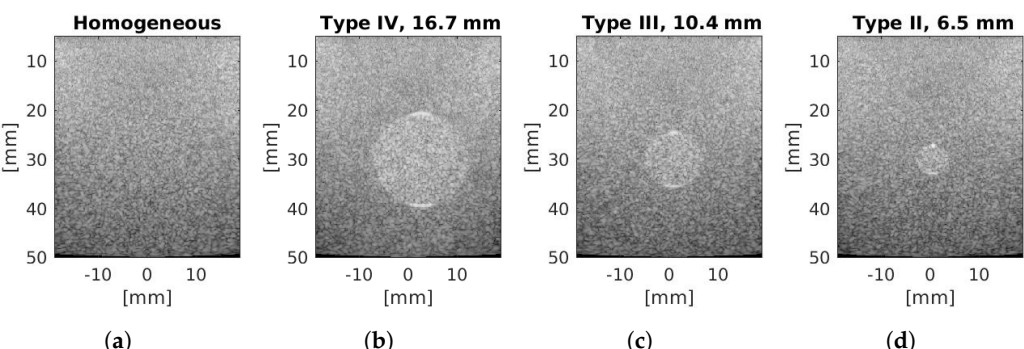

**Figure 7.** The B-mode images of the various phantom regions: (**a**) phantom region with no inclusion (background only), used in the homogeneous experiment; (**b**) type IV inclusion (70.9 kPa nominal Young's modulus and 16.7 mm diameter); (**c**) type III inclusion (34.0 kPa and 10.4 mm); (**d**) type II inclusion (8.6 kPa and 6.5 mm).

Figure 8 presents the shear wave motions in the homogeneous experiments, which were detected from the series of beamformed IQ data frames using the method described in Section 2.4. Each push beam generation method induced a shear wave field with distinct characteristics. In the FCUSE technique, there were three strong shear wave sources that generated shear wave fronts with the largest particle axial displacement due to having the highest push pulse length. However, the strong shear waves were only present at a limited

DOF, between 10–30 mm. At depths above 30 mm, the shear wave fronts were also detectable, but the particle axial velocities were smaller. The SSI technique induced broad shear wave fronts that covered most of the FOV. The lower push pulse length in the SSI method resulted in the peak axial velocity of the shear waves decreasing by an order of 5. It also caused the shear wave detection noise to become more apparent, especially at depths above 35 mm. The FCUSE-SSI method generated a wave field that combined the features of the other two techniques. Broad shear wave fronts were generated from multiple sources, which covered the majority of the FOV. In terms of shear wave peak axial velocity, the FCUSE-SSI method produced values that were lower than those from FCUSE, but higher than those from SSI.

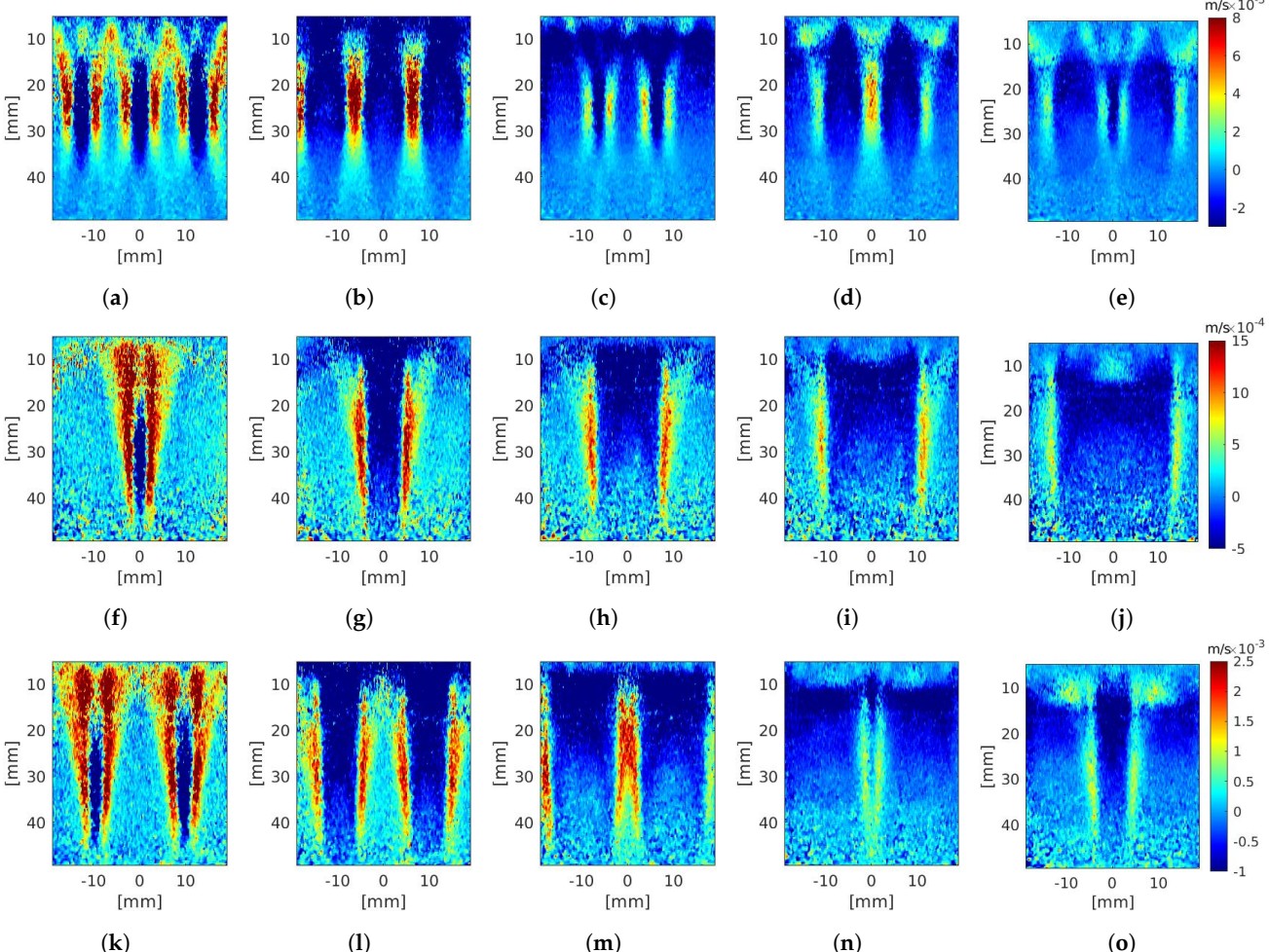

**Figure 8.** The shear wave motion data that were captured in the homogeneous experiments. Each frame shows a transient particle axial velocity map: the top row (**a–e**) shows the results from the FCUSE technique; the middle row (**f–j**) shows the results from the SSI technique; the bottom row (**k–o**) shows the results from the FCUSE-SSI technique. Note the different color map limits that were used for each technique. Images (**a,f,k**) show the first frames that were captured in each acquisition. The frame period in these images was 0.6 ms. The data after the frequency filtration are also shown.

The results from the intermediate stages of the shear wave motion data filtering in the $k - \omega$ domain are shown in Figure 9. Except for the separation of the shear waves that were propagating in opposite directions (RL and LR), the filtering in the Fourier domain (Figure 9c) allowed us to suppress the shear wave velocities that were outside of the range of interest and filter out shear waves in the low-frequency range (below 50 Hz), which improved the SNR of the shear wave motion data (Figure 9d).

Since the processing of the LR and RL datasets was performed in two independent processes, portions from two ToF-based SWS reconstructions were taken for the purpose

of compounding the final result, as shown in Figure 10 for the example of an SWS reconstruction of a type III inclusion using the FCUSE-SSI method. Each of the two SWS maps (Figure 10a,b) contained a part that primarily featured artifacts. Those regions exhibited results of low accuracy, which came from the estimation of propagation time that was found with small correlation factors (Figure 10d,e). Cutting out those regions and compounding the remaining parts of the SWS maps using Equation (7) allowed us to obtain a single SWS map (Figure 10c). In our general procedure, the compounded image was smoothed using a median filter with a manually adjusted kernel size (Figure 10f). In the remaining part of the paper, however, the compounded SWS maps are presented before median filtering in order to expose the characteristics of the various push beam generation methods without data being blurred by a filter.

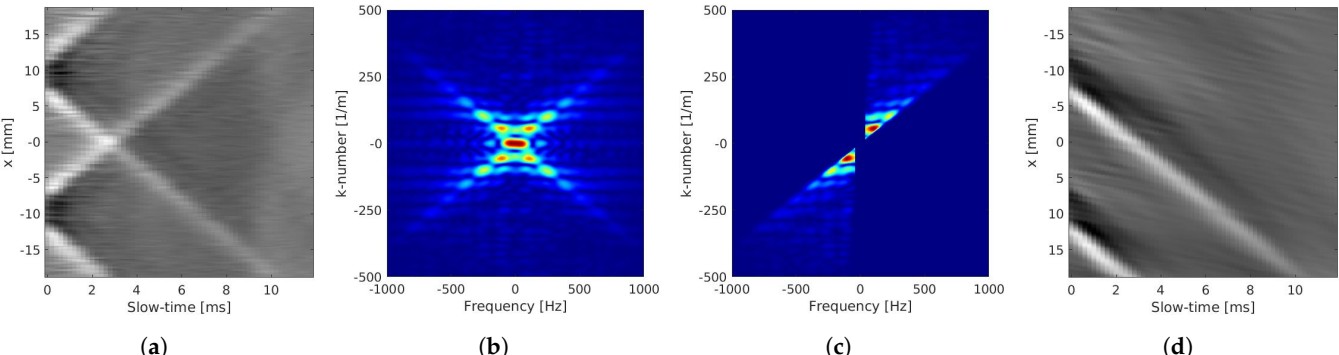

**Figure 9.** The directional filtering of the motion data in the $k - \omega$ domain: (**a**) a motion data slice from the FCUSE-SSI acquisition in the homogeneous phantom experiments from a depth of 30 mm; (**b**) the 2D FFT result of (**a**); (**c**) filtering in the $k - \omega$ domain by multiplication; (**d**) the result of the inverse 2D FFT of (**c**).

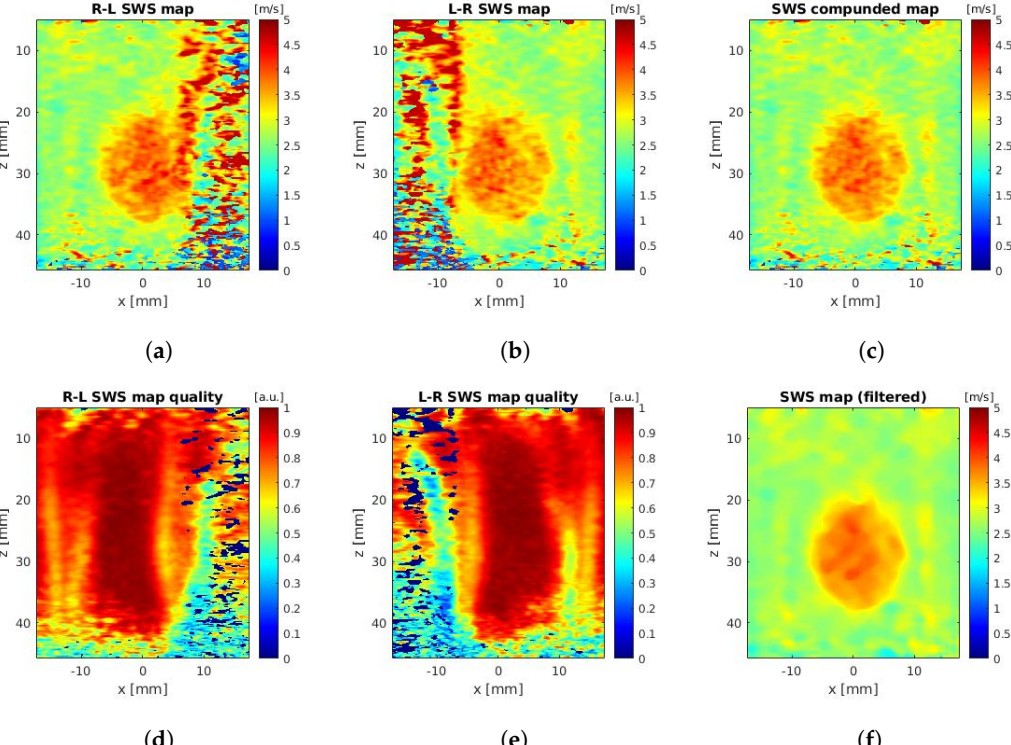

**Figure 10.** An example of a compounded SWS map using FCUSE-SSI acquisition for a type III inclusion: (**a**,**b**) the SWS maps that were reconstructed from the shear waves that were propagating from the left and right, respectively; (**d**,**e**) the quality maps for the SWS estimates in (**a**,**b**); (**c**) the compounded SWS map that was formed by combining (**a**,**b**); (**f**) the final image, i.e., the result of filtering (**c**) using a median filter with a kernel size of $2 \times 2$ mm.

### 3.2. Homogeneous Phantom Experiments

All three proposed beam generation techniques were used to acquire SWS maps of the homogeneous part of the phantom and achieved comparable results (Figure 11). The results of the ROI-based quality metric analysis (listed in Table 2) showed that the differences in the estimated average SWSs within an ROI were smaller than 0.03 m/s. Considering the nominal value of the homogeneous phantom SWS of 2.36 m/s, the average SWS estimates were biased high in all cases. In terms of the SNR, both the FCUSE and FCUSE-SSI methods showed a similar performance, while the SSI method outperformed the other techniques by producing an SNR that was higher by almost 2 dB. Furthermore, the FCUSE SWS map suffered from artifacts in the near field and close to the sides of the FOV. All techniques exhibited a significant amount of noise at depths of over 40 mm.

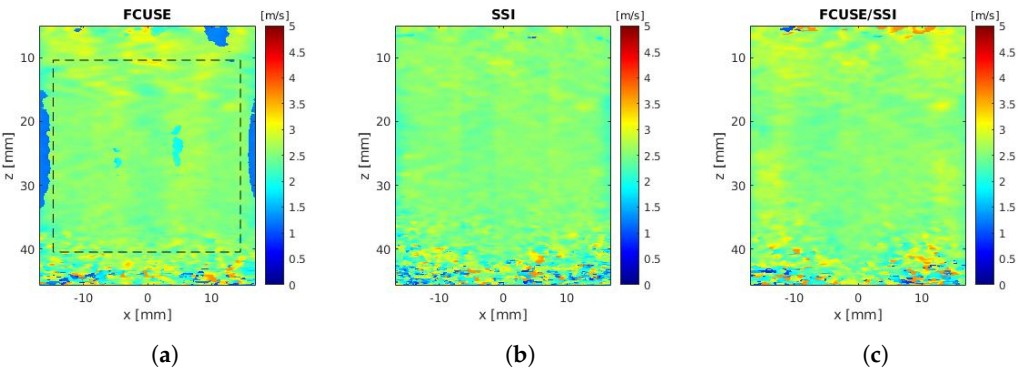

(a)        (b)        (c)

**Figure 11.** The SWS maps that were obtained in the homogeneous phantom experiments for the various beam generation techniques: (**a**) FCUSE; (**b**) SSI; (**c**) FCUSE-SSI. The ROI that was used to obtain the image quality metrics is marked in Figure (**a**). The same region was used for all techniques. The detailed results can be found in Table 2.

**Table 2.** A summary of the results from the homogeneous experiments.

| Parameter | FCUSE | SSI | FCUSE-SSI |
|---|---|---|---|
| Average SWS (m/s) | 2.56 ± 0.13 | 2.54 ± 0.10 | 2.57 ± 0.13 |
| Average bias (m/s) | 0.21 (+9.0%) | 0.18 (+7.8%) | 0.22 (+9.2%) |
| SNR (dB) | 26.0 | 27.9 | 26.2 |

### 3.3. Heterogeneous Phantom Experiments

The ability to detect inclusions with stiffnesses that were both higher and lower than that of the background was also tested using a phantom. Three out of the four available target stiffness values were chosen for these tests, with nominal Young's moduli of 8.6 kPa (type II), 34.0 kPa (type III) and 70.9 kPa (type IV), which (using Equation (2)) corresponded to nominal shear wave speeds of 1.69 m/s, 3.37 m/s and 4.86 m/s, respectively. The background material had a Young's modulus of 16.7 kPa, which mapped to an SWS of 2.36 m/s. In the experiments, target diameters of 16.7, 10.4 and 6.5 mm were used, centered at a depth of 30 mm. B-mode imaging was used to find the probe position that produced an inclusion that was centered in the FOV. In the case of the type IV inclusion, this probe placement was not possible due to the mechanical design of the phantom; thus, the inclusion was shifted slightly to the right. The results of the heterogeneous phantom experiments are presented in Figures 12–14. The detailed results are listed in Tables A1–A3 in the Appendix A. A visualization of the SWS estimates for the inclusions is shown in the form of bar plots in Figure 15.

The SWS maps that were obtained for the type IV inclusion (Figure 12) showed good contrast, especially for the inclusion with the largest diameter. The SWS estimate for the inclusion decreased along with the diameter for all of the used push generation techniques. For example, when using the SSI technique, the SWS estimate for the inclusion



dropped from 4.42 m/s to 3.38 m/s between the inclusion diameters of 16.7 and 6.5 mm, thereby dropping by nearly 24%. All of the results were biased low, by a minimum of 7.9% for inclusion size 16.7 mm to a maximum of 30.4% for smaller diameters, whereas the background SWS estimate was biased high, by 7.3% to 16.8%. The FCUSE-SSI technique suffered the highest background SWS estimate bias. Although the inclusion SWS bias was similar in all techniques, the SSI method consistently showed the lowest background SWS estimation bias compared to the other methods. The SSI method also showed the lowest background SWS estimation bias.

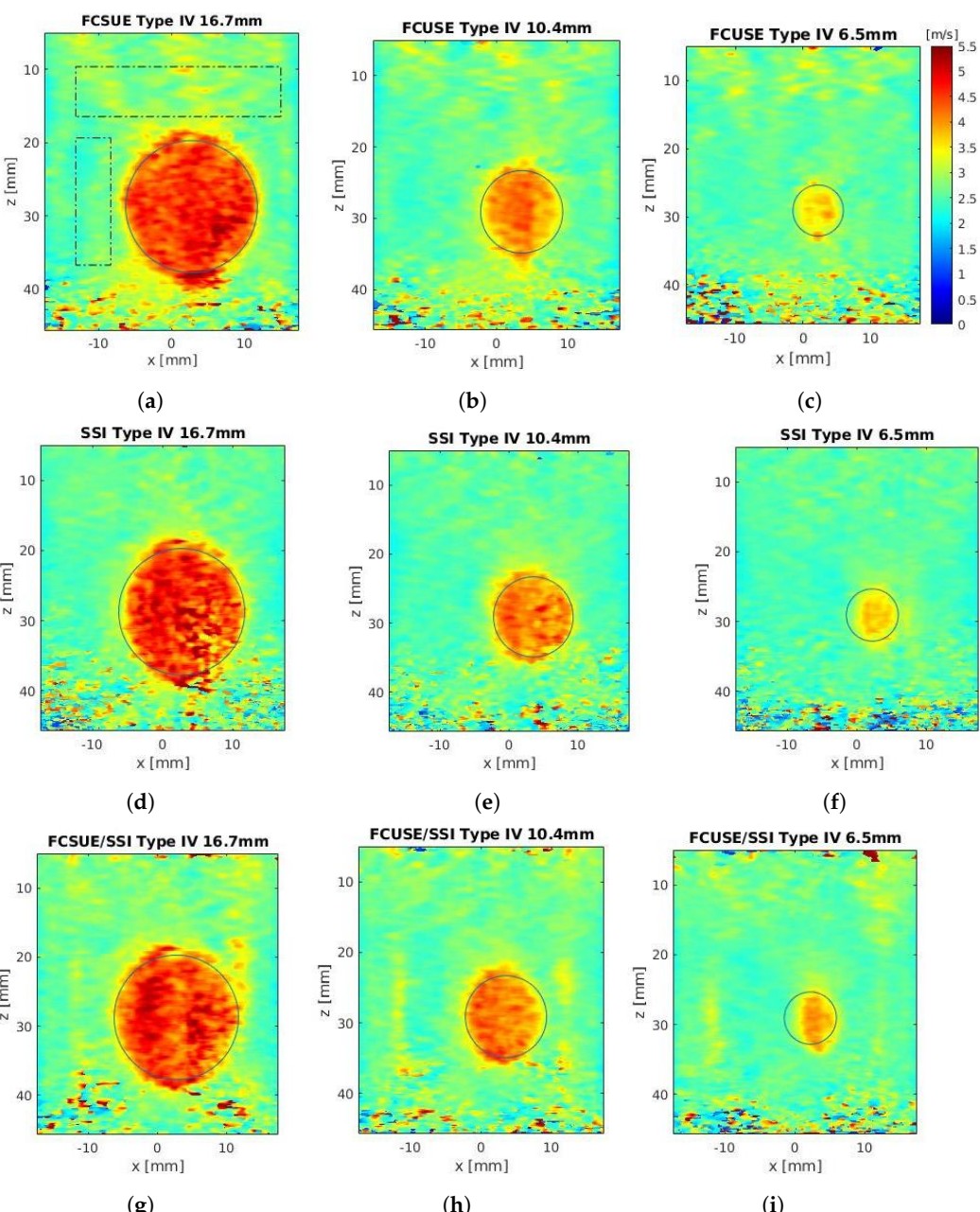

**Figure 12.** The results of the heterogeneous phantom experiments for the type IV inclusion (Young's modulus of 70.9 kPa, which corresponded to a nominal SWS of 4.86 m/s): the top row (**a**–**c**) shows the results from the FCUSE technique; the middle row (**d**–**f**) shows the results from the SSI technique; the bottom row (**g**–**i**) shows the results from the FCUSE-SSI technique. The detailed quality metrics can be found in Table A1 in the Appendix A.

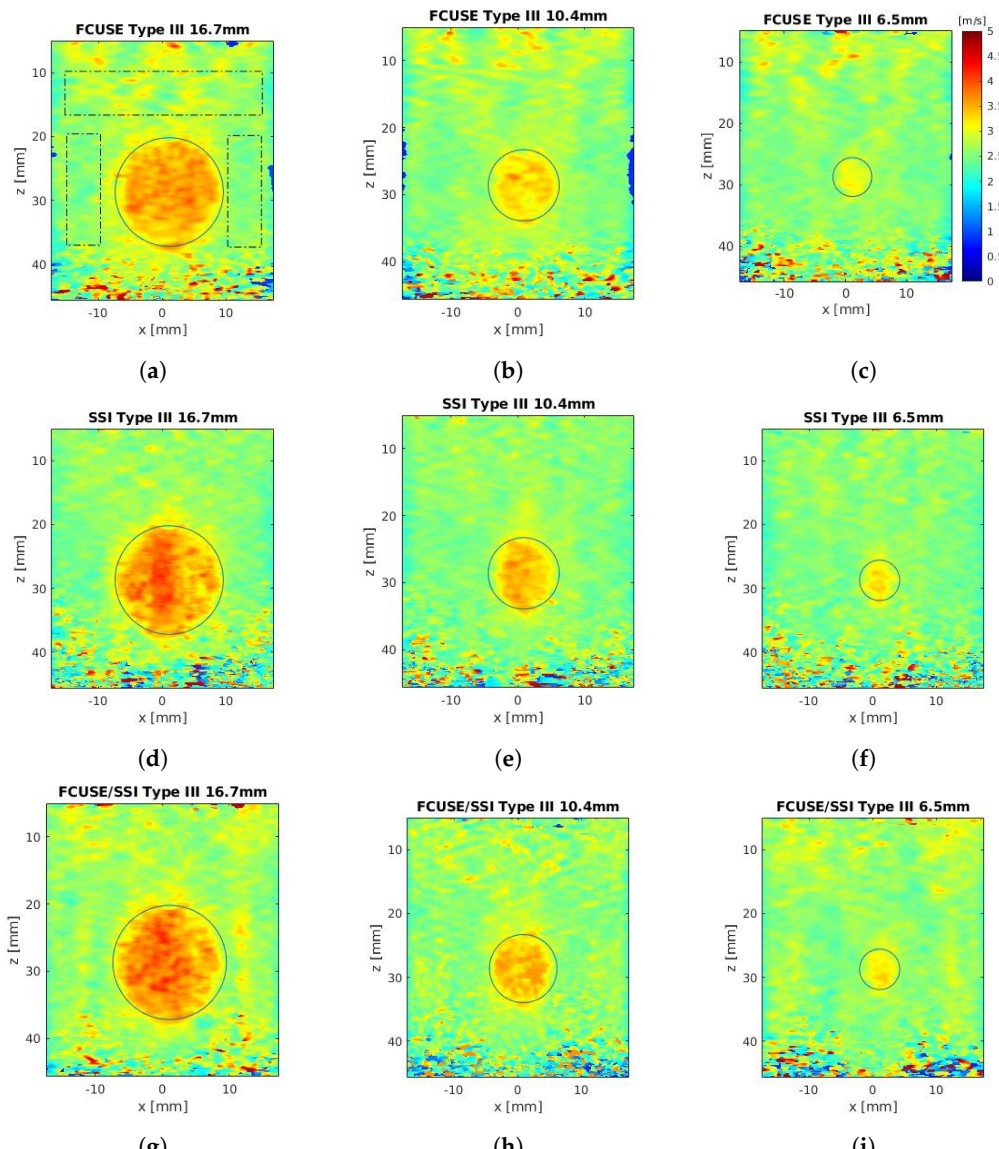

**Figure 13.** The results of the heterogeneous phantom experiment for the type III inclusion (Young's modulus of 34.0 kPa, which corresponded to a nominal SWS of 3.37 m/s): the top row (**a**–**c**) shows the results from the FCUSE technique; the middle row (**d**–**f**) shows the results from the SSI technique; the bottom row (**g**–**i**) shows the results from the FCUSE-SSI technique. The detailed quality metrics can be found in Table A2 in the Appendix A.

Due to the smaller nominal SWS difference between the inclusion and the background material, the type III inclusion SWS maps (Figure 13) showed worse contrast than those in the previous case. The same trend could also be observed, with the inclusion SWS estimates decreasing along with the inclusion diameter for all of the examined techniques. For the smallest inclusion size, this made the inclusion barely detectable in the image as the artifacts in the background were of comparable sizes and signal levels. Out of all of the techniques used, FCUSE produced the best inclusion SNR, while FCUSE-SSI produced the best background SNR. No strict trend was observed for the other characteristics.

In the case of the type II inclusion SWS maps (Figure 14), all of the inclusions could be seen easily as the low SWS value of the inclusion material did not appear in any other image region. In contrast to the previous two cases, the type II inclusion images exhibited a reverse trend of the inclusion diameter effects on the SWS estimates. Here, the inclusion SWS estimates increased with a decreasing inclusion diameter. In addition, all of the results were biased high.

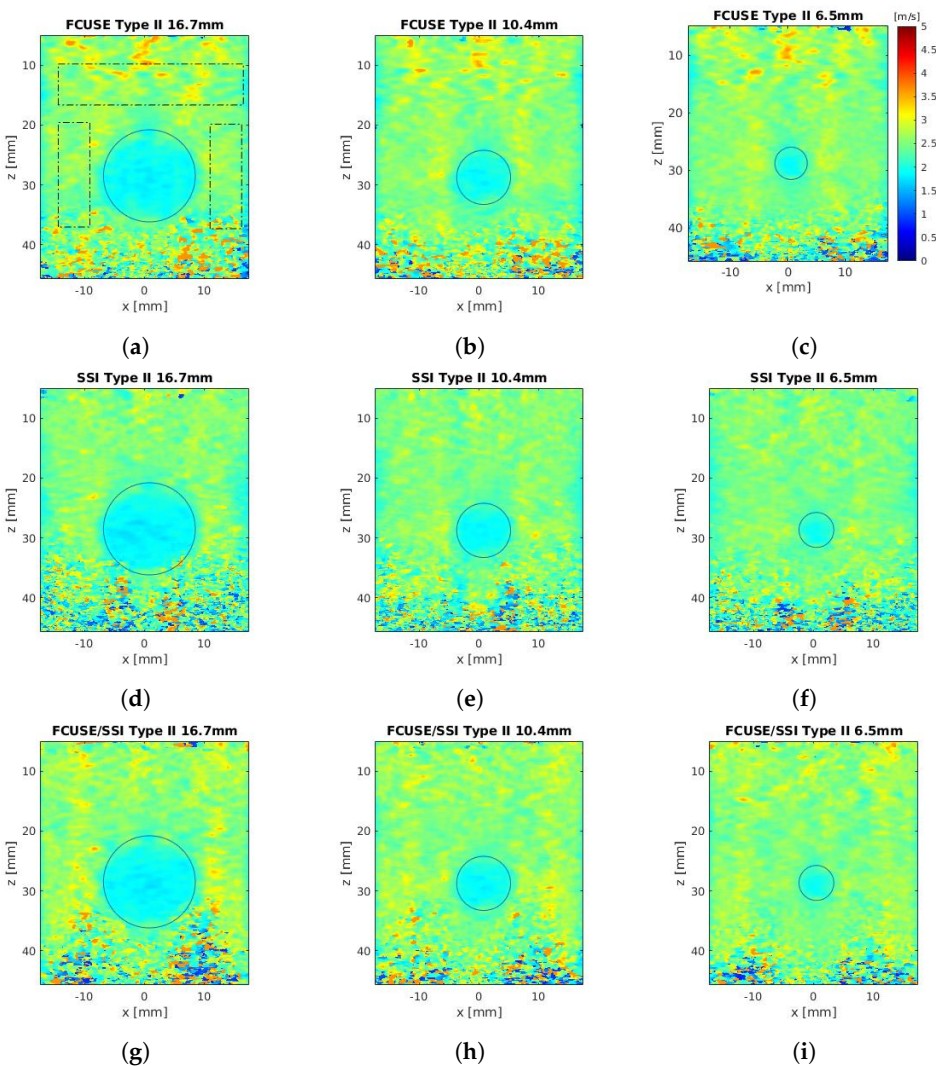

**Figure 14.** The results of the heterogeneous phantom experiment for the type II inclusion (Young's modulus of 8.6 kPa, which corresponded to a nominal SWS of 1.69 m/s): the top row (**a**–**c**) shows the results from the FCUSE technique; the middle row (**d**–**f**) shows the results from the SSI technique; the bottom row (**g**–**i**) shows the results from the FCUSE-SSI technique. The detailed quality metrics can be found in Table A3 in the Appendix A.

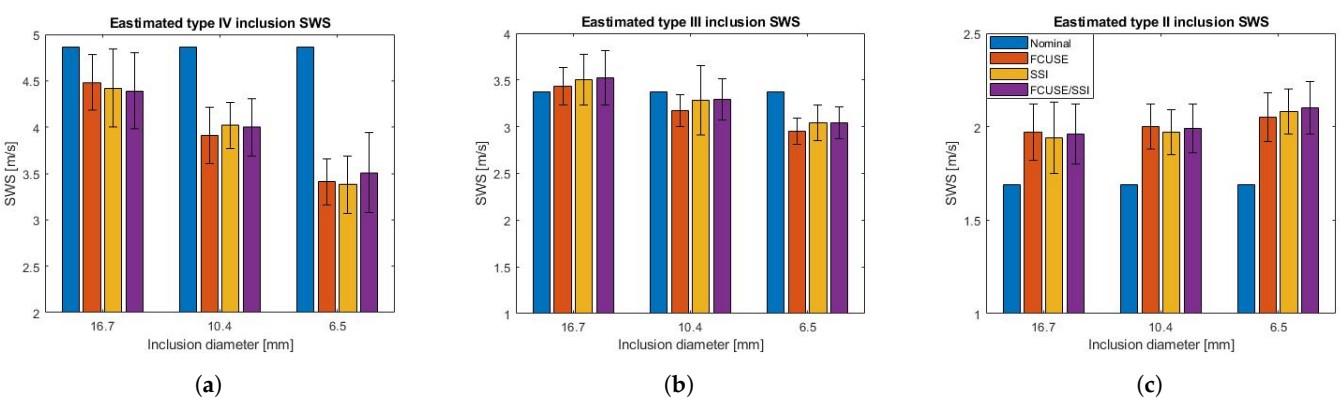

**Figure 15.** A comparison of the results from the heterogeneous phantom experiments for the various techniques and inclusion diameters: (**a**) type IV inclusion; (**b**) type III inclusion; (**c**) type II inclusion. The legend shown in figure (**c**) applies to all plots. The standard deviations of the SWS estimates within the ROIs are used as the error bars.

## 4. Discussion

### 4.1. System Validation

The presented experimental results confirmed that the proposed system was capable of implementing the SWE technique. It was achieved by satisfying two key requirements: the generation of high-energy push pulses and the high-frame-rate data acquisition of shear wave propagation. The hardware capability of the push beam generation was proven for three different methods: FCUSE, SSI and FCUSE-SSI. Each of these techniques featured some kind of complexity: parallel focused beam generation (FCUSE), the generation of a sequence of beams with strict timing (SSI) or a combination of both of these features (FCUSE-SSI). The transmit flexibility of our system showed itself to be capable of implementing all of these techniques. The second requirement for SWE was achieved by the PWI data capture with a 5 kHz frame rate. The data quality allowed us to detect the small axial displacements in shear waves with various energy levels, which were generated using the various methods (Figure 8). The correct shear wave tracking enabled full SWS map reconstruction using the ToF correlation-based SWS estimation using the developed algorithm, preceded by the 3D shear wave motion data filtering in both the time and Fourier domains to improve the conditions for the correlation calculation.

### 4.2. Phantom Experiments

In this study, two sets of phantom experiments were performed to evaluate the system performance and compare the three different push beam generation methods. In the homogeneous part of the study, all of the techniques performed similarly, even though there were significant differences between the methods, especially in terms of the energy that was transmitted by a single shear wave source. A closer look at the acquired SWS maps exposed variations in the SWS estimates, depending on the pixel position. FCUSE showed increased SWS values near the center, while these were near the sides of the FOV during the FCUSE-SSI method. The SWS estimation accuracy could depend on the local shear wave energy, as stronger displacement signals could correlate better. Because the total energy that was utilized by each technique was the same, the shear wave energy distribution, as defined by Equation (8), resulted in a different energy distribution in each technique (Figure 16). The regions of reduced energy on the push beam axis or at depths below 30 mm could be clearly seen, which was possibly the result of diffraction and attenuation. The local energy was especially high in the regions that were covered by shear waves from multiple sources. During the FCUSE method, the energy was distributed close to the focal points, which were positioned shallower than 30 mm (as in the transmit plan). This was the result of the increased $F_\#$ value of 2.4. In the SSI and FCUSE-SSI techniques, the energy was distributed over a large part of the FOV, although the energy levels were not as high as those in the FCUSE method.

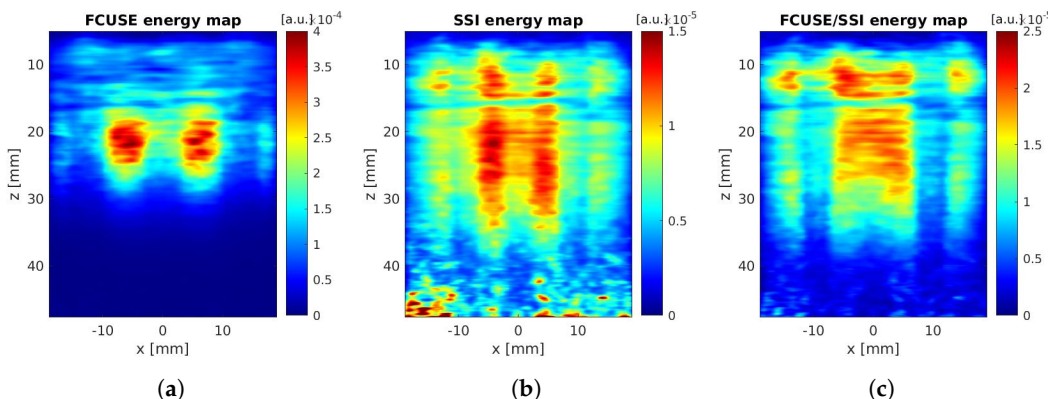

**Figure 16.** The shear wave energy distributions within the FOVs for the various beam generation methods: (**a**) FCUSE; (**b**) SSI; (**c**) FCUSE-SSI. Note the different color maps for each image.

The reconstructed SWS maps and their quality metrics confirmed that all three methods exhibited similar performances and that none of them were generally superior over the others. This was not unexpected, since the total electrical energy that was used for the push excitation was the same in all cases. The differences between the techniques could, in most cases, be explained by the differences between the shear wave energy distributions within the ROIs. Due to the lack of consistency and small differences between the quality metrics of the analyzed methods, only a simplified comparison could be conducted in this study. For inclusion SNR, the FCUSE method performed favourably compared to the other methods, showing the best results in seven out of nine measurements; however, it suffered from poor background SNR. The SSI method outperformed the other techniques in the homogeneous experiments and with regard to background bias in the heterogeneous experiments, for which it had the lowest bias in eight out of nine measurements. Finally, the FCUSE-SSI method performed well in terms of background SNR and CNR but suffered from poor inclusion SNR. It must also be noted that the comparison was not completely fair as the SSI method was the only technique that used three acquisitions to reconstruct a single SWS map, while the other techniques only used one. The push excitation energy was the same, but the SSI method required three tracking sequences instead of one and, as a result, produced a dataset for processing that was three times larger. Overall, the techniques appeared to be complimentary and could be used in conjunction with a series of acquisitions to achieve a synergy effect.

One striking result was the bias that was observed in all of the estimates. The pixel-wise SWS estimate distribution for each homogeneous phantom image (Figure 17) confirmed that in each technique, almost all pixels had values that were biased higher than the nominal value. The bias observed that was in the heterogeneous phantom experiments was even higher, ranging from −30.4% (SSI; 6.5 mm type III inclusion) to +23.8% (FCUSE-SSI; 6.5 mm type II inclusion). In general, for all techniques and all inclusion types, the absolute bias increased as the inclusion diameter became smaller. For the largest target dimension that was used (16.7 mm), the observed bias was in the range of −9.6 to −7.9% for type IV inclusions, 1.8 to 4% for type III inclusions and 14.8 to 16.4% for type II inclusions. Furthermore, it should be noted that the SWS bias resulted in an even higher bias in the elasticity maps because the Young's modulus was proportional to the square of the SWS (see Equation (2)).

In this work, the values that were provided by the phantom manufacturer were treated as the nominal values. Those values had a 5% tolerance for a given stiffness. Nevertheless, in the homogeneous case, all of the estimated average SWS values that were obtained by all three techniques were beyond this margin, which was a 2.41 m/s maximum for background material. The same applied to heterogeneous phantom results in the majority of cases. Assuming that the real SWS values were within the margins provided by the manufacturer, this led to the conclusion that there were one or more bias sources present in the system. The bias effects in the SWE technique were investigated in works by other research groups, in which it was confirmed that the SWS values that were produced with the ARF and ToF approaches depend on the transducer, aperture size and pixel position. One possible cause of bias could be an undesired push beam intensity field as wide beam widths in the elevation direction can generate significant out-of-plane shear waves that bias the SWS estimates high [29]. It has been reported that this effect is stronger when in close proximity to the push beam axis in the lateral direction and with an increasing distance from the beam focus in the axial direction. In this study, the push beam focal point depth was set close to the probe elevation focus depth (25 mm) and SWS estimates that were close to the shear wave sources were not used. Although those measures could minimize the bias, the effects could not be dismissed entirely. Diffraction could be another cause of the bias. For a push beam, diffraction produces an acoustic intensity before and beyond the depth of focus, which generates additional shear waves in non-focal regions and interferes with shear waves from the focal point, thereby biasing the results high [29]. This effect is stronger for wider apertures. Furthermore, it was assumed in the data processing

implementation that the shear waves would only propagate in the lateral direction, which was reported to be another possible reason for bias [21]. Finally, the biased results could be also caused by system-dependent parameters, such as PRF error and transducer parameter mismatches between the real values and those used in the software or phase aberration [30]. Interestingly, similar levels of bias have been reported in other studies that used the same phantom and commercial systems [31,32]. In summary, the reconstruction of SWS maps is a complex multi-step process and bias can arise at each processing stage. This then propagates through the processing pipeline, resulting in errors in the estimated SWS maps. Identifying the bias source could require a detailed analysis and was beyond the scope of this article. The same applies to the estimate variance. Exploring the effects of various system components on the bias and accuracy will be the subject of future work.

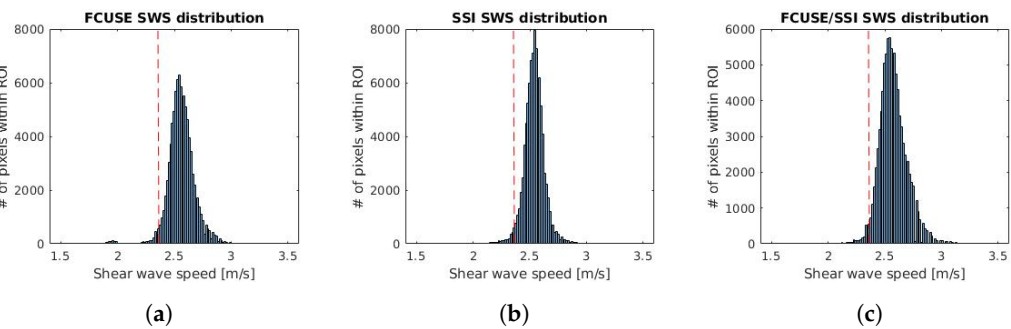

**Figure 17.** The histograms of the SWS estimates that were obtained within the ROIs in the homogeneous phantom experiments using the various beam generation techniques: (**a**) FCUSE; (**b**) SSI; (**c**) FCUSE-SSI. The red dotted line in each histogram shows the real value of the SWS (as declared by the phantom producer).

No systematic evaluation of the SWS map resolution was performed in this study. The resolution depends on many factors that are related to shear wave excitation, SWS estimation method and the parameters that are used for its calculation [6]. Importantly, there is a trade-off between the accuracy and spatial resolution that is adjusted by the lateral correlation window sizes that are used in the SWS estimation [6,28]. In this study, only a single lateral correlation window size was used (2.8 mm) in the heterogeneous experiments, which was found experimentally. In spite of that, the SWS maps of the various inclusion types and diameters (Figures 12–14) provided a brief overview of the system capabilities in terms of resolution. For instance, the 6.5 mm type III inclusion could barely be seen in the estimated SWS map due to poor contrast, whereas the type IV and type II inclusions were easily detectable. In addition, along with degrading the spatial resolution, increasing the lateral kernel size could also cause some level of averaging and shift the reconstructed inclusion values toward the SWS value of the background. This was in line with the observations of Racedo et al. [28]. This averaging effect could explain the inclusion average SWS results, which varied with respect to the inclusion diameter.

This study had some limitations. First of all, for the purpose of clarity, there was no systematic analysis of the push beam design parameters in the evaluated methods, such as focusing depth, the number of parallel beams of focal points or center frequency. Only arbitrarily applied settings were used, which might not have been fully representative of a given technique. However, an analysis of the impacts of those parameters on the SWE imaging quality has been covered in detail in other works. Secondly, only a single homogeneous phantom was used, so the evaluation of the performance of the three methods in the homogeneous experiments was limited.

*4.3. Future Directions*

In future work, we plan to address the bias that is present in the system by identifying the bias sources and minimizing their effects using system calibration and improvements in the data processing pipeline. It is crucial to enhance the overall system performance in terms of resolution and estimation accuracy, which can be achieved by increasing the robustness of the reconstruction algorithm, the optimization of the push excitation parameters or by adjusting the data acquisition scheme. Another interesting topic would be the optimization of the system performance in terms of power and the real-time implementation of the reconstructions because, for the moment, only offline reconstruction is available.

## 5. Conclusions

We presented a complete SWE implementation using a low-cost, portable and fully configurable 256TX/64RX research system. To prepare the system for exploring various SWE techniques, its transmit section performance was enhanced by introducing a new dedicated transmit extension module. In the experimental part of the study, three different push beam generation methods (FCUSE, SSI and FCUSE-SSI) were evaluated and compared in both homogeneous and heterogeneous experiments on an industry-standard elastography Q/A phantom. The achieved results for the elasticity estimations were comparable to those presented in the literature. A fair comparison of the push energy in these methods showed that the techniques were complimentary and that each had its own unique advantages and disadvantages. The developed system and processing algorithm were validated experimentally, thus proving the suitability of the proposed solution for research purposes. To our knowledge, this is the first compact and portable solution that implements the SWE technique and thus, could pave the way for new point-of-care scanners that are equipped with elasticity imaging.

**Author Contributions:** Conceptualization, D.C. and M.L.; methodology, D.C.; software, D.C.; validation, D.C. and M.L.; formal analysis, D.C.; investigation, D.C.; resources, M.L.; data curation, D.C.; writing—original draft preparation, D.C.; writing—review and editing, M.L. and D.C.; visualization, D.C.; supervision, M.L.; project administration, M.L.; funding acquisition, M.L. All authors have read and agreed to the published version of the manuscript.

**Funding:** One of the authors (D.C.) is the participant of a program by the Polish Ministry of Science and Higher Education entitled "Doktorat wdrożeniowy" (Industrial Doctoral Program). Some parts of this study were financed from the funds received under this program.

**Institutional Review Board Statement:** Not applicable.

**Informed Consent Statement:** Not applicable.

**Data Availability Statement:** Not applicable.

**Acknowledgments:** The authors would like to thank Piotr Jarosik for his assistance with the software development and Mateusz Walczak and Beata Witek for their help with the hardware development that was used in this study.

**Conflicts of Interest:** Both authors are active employees of us4us Ltd., which is a manufacturer of the research platform that was used in this study.

## Abbreviations

The following abbreviations are used in this manuscript:

| | |
|---|---|
| ADC | Analog to digital converter |
| ARF | Acoustic radiation force |
| CPWI | Compounded plane wave imaging |
| DOF | Depth of field |

| | |
|---|---|
| FFT | Fast Fourier transform |
| FOV | Field of view |
| FPGA | Field-programmable gate array |
| fps | Frames per second |
| HV | High voltage |
| IC | Integrated circuit |
| PRF | Pulse repetition frequency |
| PRI | Pulse repetition interval |
| RAM | Random access memory |
| RF | Radio frequency |
| ROI | Region of interest |
| RX | Receive |
| SWE | Shear wave elastography |
| SWS | Shear wave speed |
| ToF | Time-of-flight |
| TX | Transmit |
| TXPB | Transmit push beamformer |

## Appendix A

**Table A1.** A summary of the results from the heterogeneous experiments for the type IV inclusion (70.9 kPa and 4.86 m/s).

| Parameter | FCUSE | SSI | FCUSE-SSI |
|---|---|---|---|
| Inclusion Diameter = 16.7 mm | | | |
| Inclusion SWS (m/s) | 4.48 ± 0.30 | 4.42 ± 0.42 | 4.39 ± 0.41 |
| Inclusion bias (%) | −7.9 | −9.0 | −9.6 |
| Inclusion SNR (dB) | 23.45 | 19.53 | 20.65 |
| Background SWS (m/s) | 2.71 ± 0.24 | 2.62 ± 0.19 | 2.76 ± 0.18 |
| Background bias (%) | +14.9 | +11.2 | +16.8 |
| Background SNR (dB) | 21.23 | 22.87 | 23.57 |
| CNR (dB) | 13.3 | 11.07 | 11.27 |
| Inclusion Diameter = 10.4 mm | | | |
| Inclusion SWS (m/s) | 3.91 ± 0.30 | 4.02 ± 0.25 | 4.00 ± 0.31 |
| Inclusion bias (%) | −19.6 | −17.3 | −17.8 |
| Inclusion SNR (dB) | 22.30 | 24.02 | 22.32 |
| Background SWS (m/s) | 2.6 ± 0.19 | 2.55 ± 0.14 | 2.70 ± 0.19 |
| Background bias (%) | +10.2 | +7.9 | +14.5 |
| Background SNR (dB) | 22.51 | 25.26 | 22.99 |
| CNR (dB) | 11.28 | 14.15 | 11.09 |
| Inclusion Diameter = 6.5 mm | | | |
| Inclusion SWS (m/s) | 3.41 ± 0.25 | 3.38 ± 0.31 | 3.51 ± 0.43 |
| Inclusion bias (%) | −29.8 | −30.4 | −27.7 |
| Inclusion SNR (dB) | 22.30 | 20.90 | 18.27 |
| Background SWS (m/s) | 2.60 ± 0.19 | 2.53 ± 0.13 | 2.68 ± 0.18 |
| Background bias (%) | +10.2 | +7.3 | +13.4 |
| Background SNR (dB) | 22.51 | 25.95 | 23.50 |
| CNR (dB) | 11.28 | 8.21 | 5.13 |

**Table A2.** A summary of the results from the heterogeneous experiments for the type III inclusion (34.0 kPa and 3.37 m/s).

| Parameter | FCUSE | SSI | FCUSE-SSI |
|---|---|---|---|
| Inclusion Diameter = 16.7 mm | | | |
| Inclusion SWS (m/s) | 3.43 ± 0.20 | 3.50 ± 0.27 | 3.52 ± 0.29 |
| Inclusion bias (%) | +1.8 | +4.0 | +4.4 |
| Inclusion SNR (dB) | 24.48 | 22.12 | 21.60 |
| Background SWS (m/s) | 2.64 ± 0.20 | 2.58 ± 0.18 | 2.70 ± 0.18 |
| Background bias (%) | +12.1 | +9.3 | +14.5 |
| Background SNR (dB) | 22.38 | 23.03 | 23.74 |
| CNR (dB) | 8.73 | 8.93 | 7.55 |
| Inclusion Diameter = 10.4 mm | | | |
| Inclusion SWS (m/s) | 3.17 ± 0.17 | 3.28 ± 0.37 | 3.29 ± 0.22 |
| Inclusion bias (%) | −5.7 | −2.5 | −2.1 |
| Inclusion SNR (dB) | 25.39 | 19.06 | 23.51 |
| Background SWS (m/s) | 2.56 ± 0.24 | 2.85 ± 0.57 | 2.63 ± 0.17 |
| Background bias (%) | +8.7 | +20.80 | +11.4 |
| Background SNR (dB) | 20.59 | 13.92 | 23.63 |
| CNR (dB) | 6.32 | 3.94 | 7.52 |
| Inclusion Diameter = 6.5 mm | | | |
| Inclusion SWS (m/s) | 2.95 ± 0.14 | 3.04 ± 0.19 | 3.04 ± 0.17 |
| Inclusion bias (%) | −5.7 | −9.6 | −9.6 |
| Inclusion SNR (dB) | 25.39 | 24.24 | 24.90 |
| Background SWS (m/s) | 2.56 ± 0.24 | 2.53 ± 0.14 | 2.66 ± 0.16 |
| Background bias (%) | +8.7 | +7.4 | +12.7 |
| Background SNR (dB) | 20.59 | 24.94 | 24.62 |
| CNR (dB) | 6.32 | 6.7 | 4.24 |

**Table A3.** A summary of the results from the heterogeneous experiments results for the type II inclusion (8.6 kPa and 1.69 m/s).

| Parameter | FCUSE | SSI | FCUSE-SSI |
|---|---|---|---|
| Inclusion Diameter = 16.7 mm | | | |
| Inclusion SWS (m/s) | 1.97 ± 0.15 | 1.94 ± 0.19 | 1.96 ± 0.16 |
| Inclusion bias (%) | +16.4 | +14.8 | +15.8 |
| Inclusion SNR (dB) | 22.41 | 20.19 | 21.51 |
| Background SWS (m/s) | 2.56 ± 0.25 | 2.50 ± 0.24 | 2.64 ± 0.23 |
| Background bias (%) | +8.3 | +6.2 | +12 |
| Background SNR (dB) | 20.31 | 20.39 | 21.19 |
| CNR (dB) | 6.15 | 5.28 | 7.63 |
| Inclusion Diameter = 10.4 mm | | | |
| Inclusion SWS (m/s) | 2.00 ± 0.12 | 1.97 ± 0.12 | 1.99 ± 0.13 |
| Inclusion bias (%) | +18.3 | +16.5 | +17.4 |
| Inclusion SNR (dB) | 24.31 | 24.26 | 23.89 |
| Background SWS (m/s) | 2.55 ± 0.21 | 2.50 ± 0.19 | 2.62 ± 0.20 |
| Background bias (%) | +8.1 | +6.1 | +11.2 |
| Background SNR (dB) | 21.78 | 22.62 | 22.37 |
| CNR (dB) | 7.13 | 7.61 | 8.59 |
| Inclusion Diameter = 6.5 mm | | | |
| Inclusion SWS (m/s) | 2.05 ± 0.13 | 2.08 ± 0.12 | 2.10 ± 0.14 |
| Inclusion bias (%) | +21.0 | +22.6 | +23.80 |
| Inclusion SNR (dB) | 23.95 | 24.50 | 23.72 |
| Background SWS (m/s) | 2.55 ± 0.20 | 2.49 ± 0.17 | 2.60 ± 0.17 |
| Background bias (%) | +7.9 | +5.5 | +10.2 |
| Background SNR (dB) | 21.92 | 23.45 | 23.94 |
| CNR (dB) | 6.25 | 5.96 | 7.44 |

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
