# Peer review of "Shear Wave Elastography Implementation on a Portable Research Ultrasound System: Initial Results"

_applsci, doi:10.3390/app12126210_

Round 1
Reviewer 1 Report
The authors describe a new device for shear wave elastography measurement. After a summary of the currently available devices and an introduction of the concepts of the measurement, the authors show convincing results and good explanation of the discrepancies with the expected values.
Some of the additional work I would have like to see is mentioned as future work, which is acceptable.
One thing that would make this paper a bit stronger would be to show what results currently available commercial systems get in terms of stiffness and compare the variation to the sample manufacturer data between those system and the news system presented here. However, I recognize that this would require access to one of those expensive systems and might not be easily done.
From all this, I think this paper is worth publishing as is.
Reviewer 2 Report
Comments and Recomendations:
1. All formulas in the article are numbered. But almost none of them are mentioned in the text. It's not clear why formulas should be numbered if they are not referenced in the article.
2. Any abbreviations found in the article should be accompanied by the necessary decipherment or explanation with reference to works where such an explanation is given even if such an abbreviation is well known to the authors. For example, RF.
3. The authors give a long series of complex formulas, without bothering to explain where all this comes from, why it is necessary to write these formulas, in what conditions they are valid and when their application is impossible. The question is: why give formulas if they are not analyzed, no one tries to make estimates on their basis, simple examples do not demonstrate their physical essence? Maybe it's better to explain what their form tells to a narrow circle of initiates? Or do you do without them?
4. The volume occupied by the Fig. 12-14, in no way comparable to the size of the text that is devoted to their discussion. Perhaps it makes sense to think about how to present the information contained in the figures in a more compact form?
This paper is well enough written to understand main results. The manuscript seems to be suitable for publication. I am therefore convinced that such a work corresponds to the content of the Journal of Applied Sciences and can be published there after minor mentioned corrections.
